# Implementation of the EIRA 3 Intervention by Targeting Primary Health Care Practitioners: Effectiveness in Increasing Physical Activity

**DOI:** 10.3390/ijerph181910537

**Published:** 2021-10-08

**Authors:** Sara Contreras-Martos, Alfonso Leiva, Álvaro Sanchez, Emma Motrico, Juan Bellón, Susana Aldecoa Landesa, Rosa Magallón-Botaya, Marc Casajuana-Closas, Edurne Zabaleta-del-Olmo, Bonaventura Bolíbar, José-Ángel Maderuelo, Joan Llobera

**Affiliations:** 1Primary Care Research Unit Majorca, C/Escola Graduada 3, 07002 Palma, Spain; sara.contreras.martos@gmail.com (S.C.-M.); jllobera@ibsalut.es (J.L.); 2Balearic Islands Health Research Institute (IdISBa), 07120 Palma, Spain; 3Primary Care Research Unit of Bizkaia, Basque Health Service (BHS), Luis Power 18, 48014 Bilbao, Spain; Alvaro.Sanchez@osakidetza.net; 4Department of Psychology, Universidad Loyola Andalucia, Avda. de las Universidades s/n, 41704 Sevilla, Spain; emotrico@uloyola.es; 5Biomedical Research Institute of Málaga (IBIMA), 29010 Málaga, Spain; jabellon@uma.es; 6El Palo Health Centre, Andalusian Health Service (SAS), 29018 Málaga, Spain; 7Department of Public Health and Psychiatry, University of Málaga (UMA), 29016 Málaga, Spain; 8I-Saúde Group, Galicia Sur Health Research Institute (IIS Galicia Sur), SERGAS-UVIGO, Avda Rosalía Castro 21, 36201 Vigo, Spain; susialdecoa@gmail.com; 9Biomedical Research Institute of Aragón (IISA), Avda. San Juan Bosco 13, 50009 Zaragoza, Spain; med000764@gmail.com; 10Department of Medicine, Universidad de Zaragoza, C/ Domingo Miral 5, 50009 Zaragoza, Spain; 11Institut Universitari d’Investigació en Atenció Primària Jordi Gol (IDIAP Jordi Gol), Gran Via Corts Catalanes 587 àtic, 08007 Barcelona, Spain; mcasajuana@idiapjgol.info (M.C.-C.); ezabaleta@idiapjgol.org (E.Z.-d.-O.); bbolibar@idiapjgol.org (B.B.); 12Universitat Autònoma de Barcelona, 08193 Barcelona, Spain; 13Gerència Territorial de Barcelona, Institut Català de la Salut, c/Balmes 22, 08007 Barcelona, Spain; 14Faculty of Nursing, Universitat de Girona, Carrer d’Emili Grahit, 77, 17003 Girona, Spain; 15Primary Health Care Research Unit, Institute of Biomedical Research of Salamanca (IBSAL), Paseo de San Vicente, 58-182, 37007 Salamanca, Spain; jmaderuelo@saludcastillayleon.es; 16La Alamedilla Health Center, Health Service of Castilla y León (SACyL), Avda. Comuneros 27-31, 37003 Salamanca, Spain

**Keywords:** physical activity, multiple health behavior change intervention, primary care

## Abstract

The World Health Organization (WHO) estimated that physical inactivity (PI) is responsible for 20 to 30% of all non-communicable diseases. We aimed to analyze the effectiveness of a multiple health behavior change (MHBC) intervention to increase physical activity (PA) in patients 45 to 75 years old who had at least 2 of 3 unhealthy behaviors (tobacco use, reduced fruit and vegetable consumption, and insufficient PA). The MHBC intervention is based on the Transtheoretical Model and the conceptual framework of the “5 A’s” and includes an individually tailored intervention, group sessions, and the use of community resources. We included 3062 participants, 1481 in the intervention group and 1581 in the control group. After 12 months, there were no differences in PA intensity measured by metabolic_equivalent_of_task_minutes/week (adjusted mean difference: 284.093, 95% CI: −298.24, 866.42) nor in the proportion of participants who increased PA levels to moderate or high (OR: 1.02, 95% CI: 0.85, 1.23; *p* = 0.822), and no differences in blood pressure, weight loss, or waist circumference. We found an increased proportion of patients in the intervention group who followed the WHO recommendations for PA (OR: 1.29; 95% CI: 1.04, 1.60; *p* = 0.02). We concluded that the intervention did not lead to a significant increase in PA.

## 1. Introduction

Non-communicable diseases (NCDs) are the leading cause of mortality worldwide and are responsible for 15 million premature deaths each year. Cardiovascular diseases, cancer, and diabetes account for 80% of these premature deaths. Modifiable behaviors, such as poor diet, inadequate physical activity (PI), and the use of tobacco and alcohol, increase the risk for most NCDs [1].

The pioneering 1954 study by Sir Alan Rook clearly demonstrated the health benefits of physical activity (PA) [2]. More recent studies reported that regular PA is associated with improvements in inflammatory/hemostatic biomarkers, lipoprotein profile, hemoglobin A1c, homocysteine, and serum creatinine, and also aids in blood pressure control and weight management [3,4]. Engagement in PA decreases with age, with sedentary behaviors being highly prevalent in older adults [5,6]. The increase in sedentary behaviors and the concurrent decrease in PA are positively associated with increased incidences of obesity and hypertension [7,8,9]. Greater PA is associated with decreases in blood pressure, body mass index, and lipid concentrations [10].

The World Health Organization (WHO) estimated that physical inactivity was responsible for 20 to 30% of all NCDs and that about one-third of the adult population did not engage in sufficient PA [11]. Considering the health benefits of PA and the global burden of NCDs, the promotion of PA is a major public health priority. The WHO and the Global Observatory for Physical Activity (GoPA) have integrated the efforts of governments and stakeholders to develop new policies and strategies that reduce the global prevalence of insufficient PA by 10% among adults in 2025 [12,13].

The ‘Physical activity strategy for the WHO European Region 2016–25’ recognized the importance of contributions by primary care practitioners in achieving its goals. In particular, it recommended that ‘... making the promotion of physical activity by health professionals the norm’ and that ‘early identification, counselling and referral at the primary care level should be integrated into standard practice’ [14].

The broad coverage of primary health care (PHC) centers and their influence on population behaviors make them ideally suited to identify people at risk for NCDs and to encourage them to increase their PA [15,16]. Previous research reported that interventions aimed at promoting PA in primary care settings had modest but positive effects [4,17,18,19], although no clear strategies have yet been developed to maximize the effectiveness of these interventions [20].

A PHC physician usually has to care for individuals who have more than one unhealthy lifestyle or behavior. Thus, the approach of multiple health behavior change (MHBC) interventions, in which more than one aspect is simultaneously addressed, was recently developed to increase the efficiency and health benefits and reduce the costs of these interventions [20,21]. However, there is little evidence of the effectiveness of MCBH interventions on increasing PA [20].

The EIRA intervention is a complex, multifaceted intervention that aims to reduce tobacco use, increase adherence to a Mediterranean diet, and increase PA. The intervention is personalized and targets individual behaviors and stages of change. The aim of this secondary study was to assess the effectiveness of an MHBC intervention in increasing PA and then evaluate the potential health benefits after 12 months.

## 2. Materials and Methods

### 2.1. Design

This study was a two-arm clustered randomized controlled clinical trial with an effectiveness-implementation hybrid design (type-2) (ClinicalTrials.gov (NCT03136211)). The purpose of the MHBC intervention was to increase PA at 3 different levels—the individual, the group, and the community—and was delivered in PHC settings. The results of this trial are reported according to the Standard for Reporting Implementation Studies and Consolidation Standards of Reporting Trials (CONSORT) statements for clustered randomized trials. The detailed protocol has been previously described [22].

### 2.2. Setting

#### 2.2.1. PHC Centers

This study was performed in 26 PHC centers in 7 of the 17 Spanish Autonomous Communities: Andalusia, Aragon, the Balearic Islands, Basque Country, Castile and León, Catalonia, and Galicia. All health care professionals and administrative staff were invited to participate. Each participating professional signed a document of collaboration indicating a commitment to the study.

#### 2.2.2. Participants

Participants from the participating PHC centers were 45 to 75 years old and had at least 2 of the following 3 unhealthy behaviors:

Tobacco use: Participants reported smoking more than one cigarette per day at the initial visit.

Physical inactivity: A PHC professional assessed PA using the Brief Physical Activity Assessment Tool [23,24], which has been used efficiently in routine primary health care services to identify insufficiently active patients who may need physical activity advice and consists of two questions: “(A) How many times a week do you usually do 20 min of vigorous PA that makes you sweat or puff and pant? (for example, jogging, heavy lifting, digging, aerobics, or fast bicycling) and (B) How many times a week do you usually do 30 min of moderate PA or walking that increases your heart rate or makes you breathe harder than normal? (for example, mowing the lawn, carrying light loads, bicycling at a regular pace, or playing doubles tennis)”. Participants were considered inactive if they reported 2 or fewer 20 min sessions of vigorous-intensity PA per week, or 4 or fewer 30 min sessions of moderate-intensity PA (including walking), or 4 or fewer sessions of any combination of moderate- and vigorous-intensity PA.

Reduced fruit and vegetable consumption: participants were asked how many servings of fruits or vegetables they ate each day. Consumption of fewer than 5 servings of vegetables or fruits per day was considered unhealthy.

Participants were excluded if they had an advanced serious illness, cognitive impairment, dependence on others for the completion of basic everyday activities, severe mental illness, involvement in a long-term home health care program, treatment for cancer or end-of-life care, or if they did not plan to reside in the area throughout the study period.

#### 2.2.3. Assignment of PHC Centers to Different Interventions

The allocation schedule for random assignment of interventions to the different PHC centers was computer-generated at a central location (IDIAP Jordi Gol, Barcelona, Spain) and stratified by region (Andalusia, Aragon, the Balearic Islands, Basque Country, Castile and León, Catalonia, and Galicia) in a 1:1 ratio. The allocation to the intervention group or usual care group was not concealed at the PHC centers. Participants and health care professionals were not blinded to the intervention.

#### 2.2.4. Intervention Group

The intervention was based on the Transtheoretical Model (TTM) [25,26] of behavior change. The TTM is an ‘integrative theory that assesses an individual’s readiness to act on a new healthier behavior, and provides strategies or processes of change to guide the individual’. The intervention, which was designed to encourage participants to move forward through successive stages (pre-contemplation, contemplation, preparation, action, maintenance, and termination), was implemented by physicians and nurses in routine PHC practices according to the conceptual framework of the “5 A’s”: Assess, Advise, Agree, Assist, and Arrange follow-up [27]. The personal level of behavior and stage of change was assessed by a PHC professional during the first screening visit (“Assess”). The TTM is a tool used by PHC professionals to guide them during the intervention and to help them decide on potential prescriptions, depending on the stage of change and the unhealthy behaviors that needed to be targeted of each participant. At that time, participants were asked one question about tobacco use during the previous month, two questions about the daily consumption of fruits and vegetables [28], and the Brief Physical Activity Assessment Tool was administered [23,24]. The intervention was tailored depending on the target behaviors and the participant stages of change for these particular behaviors using the core constructs of the TTM [25] (Appendix A). Subsequently, PHC professionals provided information and advised the person (“Advise”) until the subject agreed on a specific goal (“Agree”). The PHC professionals assisted participants in establishing and following an action plan (“Assist”) and arranged follow-up support (“Arrange”). The intervention was developed based on the results of previous preclinical studies and phase I and II clinical trials [20,29,30,31,32,33,34,35]. Priority actions to address one or more of the three unhealthy behaviors simultaneously were set up by the PHC professional together with the participant.

Normally this individual part of the intervention is tailored by PHC professionals to include up to 3 visits. After evaluating the state of change, the PHC professionals recommended either (a) a “very brief intervention” with the aim of increasing awareness and highlighting the importance of the need for behavior change or to support the change and help with relapse prevention; or (b) a “brief intervention” which included the formulation and agreement of an action plan for behavior change, preferably PA involving group sessions and community resources offered by the EIRA study. PHCs were previously trained during a 20-h in-person group feedback session and a coded acting patient session to reinforce their skills in motivational interviewing.

Before the study onset, the PHC teams identified community health assets and agreed on the referral of participants. These activities were developed for promoting physical activity and healthy food options. The community health assets for physical activity included instructor-led sessions in sports centers or outdoor activities coordinated by private or public organizations (Appendix A). Follow-up visits were performed during the first and third months after starting the PA plan for assessment and establishment of new plans if needed.

Group sessions at the PHC centers lasted 90–120 min and were led by PHC professionals at the health center. The group sessions to reinforce the recommendations provided to each individual consisted of talks that encouraged participants to engage in active lifestyles. These sessions provided guidelines to facilitate and engage participants in the practice of PA and the adoption of a healthy diet through various activities, including PA sessions, cooking workshops, and preparations of seasonal menus.

Information and communication technologies were available to support participants throughout the intervention. Participants were encouraged to install and use the EVIDENT smartphone application (App), which provides personalized recommendations and specific objectives and goals for PA. EVIDENT is a simple and user-friendly app that monitors the daily consumption of food and daily PA performed. It includes a pedometer that allows the inclusion of additional PA information, if necessary. After a daily analysis, the app generates a plan for the following days aimed at improving eating habits and increasing PA [36]. In addition, personalized SMS were periodically sent to each participant to motivate and engage them to follow the plan established with the PHC professional (including PA, tobacco, and/or diet) and a web page containing information about healthy lifestyles addressed to the participant was available (http://proyectoeira.rediapp.es, accessed on 20 November 2020).

#### 2.2.5. Usual Care Group

PHC professionals who cared for participants in the usual care group integrated the recommendations of the Program of Preventive Activities and Health Promotion into their practices [37]. This program incorporates preventive protocols that include lifestyle recommendations and a set of preventive activities that consider age, sex, and risk factors. Preventive activities were based on systematic screening and brief advice for the prevention of cardiovascular and mental diseases.

### 2.3. Outcomes

#### 2.3.1. Primary Outcome

PA was determined by measurements of the metabolic equivalent of task minutes per week (MET min/week) at 12 months. These measurements were according to the ‘Guidelines for the data processing and analysis of the international physical activity questionnaire’ (http://www.ipaq.ki.se/scoring.pdf, accessed on 20 November 2020).

#### 2.3.2. Secondary Outcomes

Secondary outcomes included PA levels classified according to the International Physical Activity (IPAQ) 7 questionnaire as low, moderate, or high [38]. The percentage of participants who followed WHO recommendations for PA was recorded. These recommendations are ‘at least 150 min of moderate-intensity PA throughout the 5-day week, or at least 75 min of vigorous-intensity PA throughout the 5-day week, or an equivalent combination of moderate- and vigorous-intensity activity’ [14]. The minutes per day of moderate- and/or high-intensity PA and time spent sitting (min/day) were also assessed. Additional secondary outcomes included: blood pressure at 12 months in mmHg measured in the upper arm using a validated monitor. PHC practitioners were instructed to measure blood pressure with the subject in a seated position after at least 5 min of rest. Waist circumference at 12 months was measured midway between the lowest rib and the iliac crest. Body mass index in kg/m^2^ was determined from measurements of the subject’s height and weight. Cardiovascular risk at 12 months was assessed using the REGICOR risk score [39,40]. Other clinical parameters, such as low-density lipoprotein (LDL), high-density lipoprotein (HDL), triglycerides (TG), glucose, and glycated hemoglobin concentrations, were evaluated at 12 months by testing a blood sample obtained during the first awake hour in the morning under fasting conditions. This score is an adaptation of the Framingham coronary risk score for the European Mediterranean area. Based on this score, participants were stratified as having low/moderate or high cardiovascular risk.

### 2.4. Sample Size

The sample size was calculated so that the study was sufficiently powered to detect differences in the proportion of participants who changed at least one unhealthy behavior [21]. A total of 3640 participants were initially recruited, and 3062 participants were finally included. The standard deviation of the MET mins/week in the study population was 2700. Assuming an alpha risk of 5%, a beta risk of 20%, and an intra-cluster correlation of 0.01 [41], this allowed detection of an effect size of 0.26 (Cohen’s d) in PA measured by MET mins/week difference between the control and intervention groups.

### 2.5. Statistical Analysis

The baseline characteristics of all participants in the control and intervention groups were recorded, and variables were expressed as frequencies or percentages (categorical variable) or as means and standard deviations (continuous variables).

Intention to treat (ITT) analysis was used to determine differences of the groups in PA (MET mins/week), weight, body mass index (BMI), waist circumference, blood pressure, LDL, HDL, TG, glucose, and glycated hemoglobin by fitting to a generalized estimation equation (GEE) model with family = Gaussian, link = identity, and correlation = exchangeable. A GEE model with family = Poisson and link = log was used to determine differences among groups in moderate- and high-intensity PA, and time spent sitting (mins/day). To analyze differences in the proportions of participants who performed moderate/high PA and who followed the WHO recommendations for PA (>150 min/week), a GEE model with family = binomial and link = logit was used. Odds ratios (OR), adjusted mean differences, and 95% confidence intervals (CIs) were reported. To assess potential biases due to incomplete follow-up and nonresponse in surveys, multiple imputations by chained equations (MICE function in R software) with 50 imputed datasets were performed. Estimates from each imputed dataset were combined using procedures outlined by Rubin [42,43]. Sensitivity analyses were used to examine the robustness of the results using different approaches (per-protocol analysis and subgroup analysis) of participants who received the intervention.

## 3. Results

We randomly allocated the 26 PHC centers that met the inclusion criteria to the intervention group (IG; *n* = 13) or the control group (CG; *n* = 13). Prior to the study onset, one of the PHC centers in the IG decided not to participate (Figure 1).

A total of 3062 participants from the 25 PHC centers met the eligibility criteria. The IG had 1481 participants, and we classified 1345 of them (90.8%) as inactive, based on the Brief Physical Activity Assessment Tool. Of these, 502 participants smoked, 1248 consumed an unhealthy diet, and 405 were at risk for the three unhealthy behaviors. Of the 1345 participants classified as inactive, 753 (56.0%) received the PA intervention depending on the personal stage of change. Of these 753 participants, 526 received individualized interventions, 75 participated in group sessions, and 177 were prescribed community activities. The CG had 1581 participants, with 1448 (91.6%) classified as inactive; these individuals were managed according to the usual care practice established in each health center. Among the 1448 inactive participants in the CG, 564 were smokers, 1349 consumed an unhealthy diet, and 465 followed all three unhealthy behaviors. Twelve months later (at the end of the study), 374 participants in the IG (25%) and 296 in the CG (19%) were lost to follow-up.

We initially compared the baseline characteristics of participants in the IG and CG (Table 1).

The IG had a higher BMI, a lower PA, and more participants in the ‘preparation state’ (40.6% vs. 13.2%). Slightly more than half of the participants in each group were women (IG: 55.2%, CG: 54.6%), and the groups had a similar mean age (IG: 57.7 years ± 7.9; 95% CI: 57.3, 58.1; CG: 58.3 years ± 8.3; 95% CI: 57.9, 58.7). Almost half of the study population were smokers (IG: 43.3%, CG: 44.1%), and most participants did not follow a healthy diet (consumption of at least five servings of fruits or vegetables per day; IG: 83.8%; CG: 80.0%). About 40% of participants had hypertension (IG: 40.2%; CG: 38.7%) and 6% had a cardiovascular disease (IG: 6%; CG: 5.5%).

PA data were available from 2392 participants, 1285 in the CG and 1107 in the IG. Differences in adjusted mean change from baseline between intervention versus control at 12 months are presented in Table 2.

After 12 months, these groups did not differ significantly in PA intensity (284.093 MET min/week; 95% CI: −298.24, 866.42; *p* = 0.335), the time spent on high-intensity PA (0.15 min/day; 95% CI: −0.20, 0.49; *p* = 0.402), and the time spent sitting (−0.003 min/day; 95% CI: −0.07, 0.06; *p* = 0.920). In the per-protocol analysis, the IG showed a significant but slight increase in the time spent performing moderate-intensity PA (0.25 min/day; 95% CI: 0.01, 0.49; *p* = 0.038), but this difference was not significant on ITT analysis. These groups also had no difference in the proportion of participants who increased the level of PA to moderate or high (OR: 1.02; 95% CI: 0.85, 1.23; *p* = 0.822). However, significantly more participants in the IG followed the WHO recommendations regarding PA (OR: 1.29; 95% CI: 1.04, 1.60; *p* = 0.02).

We also determined the effect of the intervention on blood pressure and cardiovascular risk. The IG had a significantly lower systolic blood pressure (−1.96 mmHg; 95% CI: −3.13, −0.79; *p* = 0.001) and cardiovascular risk score (−0.26; 95% CI: −0.5, −0.03; *p* = 0.030) in the per-protocol analysis, but these differences were not significant in the ITT analysis. In addition, per-protocol analysis showed that the IG had a significantly higher waist circumference at 12 months (0.60 cm; 95% CI: 0.02, 1.18; *p* = 0.043), but this difference was not significant on ITT analysis. The two groups also showed no significant differences in weight loss and in LDL, HDL, TG, glucose, and glycated hemoglobin concentrations.

## 4. Discussion

Our study examined the effectiveness of the MHBC intervention in adults who had at least 2 of 3 unhealthy behaviors and attended PHC centers in different regions of Spain. The results indicated that the intervention led to only a small increase in PA, in that it slightly increased the number of participants who followed the WHO recommendations for PA at the end of the 12-month period. However, the IG and CG had no differences in total PA (MET mins/week) and PA intensity. These groups also did not differ in blood pressure, weight loss, waist circumference, or in the concentrations of LDL, HDL, TG, glucose, and glycated hemoglobin after 12 months.

The health benefits of an active lifestyle and sustained PA are widely acknowledged by the scientific community and health professionals, and there is, therefore, a great interest in developing efficient methods to encourage the general population to engage in PA [14]. However, the success of promoting PA in primary care settings requires PHC professionals to accept this additional task of efficiently promoting PA. There is evidence that PA promotion in the primary care setting is effective but only has a moderately positive effect (range of RR: 1.22 to 1.42) and a high degree of heterogeneity (range of I2: 67 to 83.5%), indicating substantial variation among studies [17,20]. This diversity of results may be related to differences in referral procedures, programs, and the characteristics of the intervention [18,44]. In the present study, the intervention differed among participants in the IG because it was modified according to the characteristics of each individual. The intervention depended on the participant’s stage of change, unhealthy behavior and availability of community resources. In addition, the PHC professional could prescribe one or more different types of interventions (individual sessions, group sessions, and referral to community resources). Only about half of the individuals in the IG received at least one of these three types of interventions. Moreover, the implementation of the interventions was challenging, and the execution was not always successful. Further studies should reconsider the strategies for the implementation of PA interventions to overcome potential barriers faced by PHC professionals. For example, there should be a focus on simpler methods for the more uniform prescription of PA. Overall, it is likely that variations in the intervention used in the present study influenced their effectiveness and hindered their evaluation.

Our EIRA trial showed that the MHBC intervention led to no significant improvements in PA or clinical biomarkers at 12 months. In contrast, some other studies found that a minimal intervention increased PA [45,46,47] but had no impact on other objective measures of health (weight and blood pressure). A study of the ‘Beweegkuur’ intervention in the Netherlands reported it was effective in increasing mobility, improving mood, and reducing pain, blood pressure, blood sugar, and body weight [48]. The 2017 “Pas-a-pas” study in Spain, a community intervention program that aimed to increase PA and socio-cultural activities, also reported positive results [49]. This Spanish study reported that PA activity was significantly greater even 2 years after the intervention and that the intervention group had reductions in systolic blood pressure, total cholesterol, and LDL-cholesterol, but no significant differences in body weight, BMI, or waist circumference [48]. It is worth noting that our MHBC intervention simultaneously targeted physical inactivity and unhealthy eating behaviors of participants presenting with those risks. The obtained results for LDL, HDL, TG, blood sugar, and glycated hemoglobin may have been influenced by both the PA and dietary components of the intervention. Thus, the differences in these clinical parameters may not be due solely to the effects of PA. Moreover, promotion of a Mediterranean-based diet, with no restrictions in the consumption of fruits, olive oil, nuts, and other products rich in natural sugars and heart-healthy fats, may have also played a role in some of the outcomes (for some of the participants) at 12 months.

The most effective intervention for promoting PA is unknown. Campbell et al. examined a PA intervention and was unable to identify features of the intervention content, setting, or delivery that led to better outcomes and also reported no clear benefit from a more extensive intervention [50]. Other authors found that brief interventions, consisting of a few minutes of advice, sometimes including referral for further interventions, increased PA compared to usual care. However, there is insufficient evidence to conclude whether exercise referral or advice and counseling is more effective [45]. PA programs involving a specialist, social support, and social participation seem to produce small positive effects in maintaining regular PA, and combination with individual practice, community-based activities, and group sessions improves long-term adherence [51]. Our MHBC intervention incorporated several of these elements in that some individuals received brief individual interventions from a health professional and referral to group sessions and community activities that were directed in some cases by specialists or instructors. However, this complete intervention was not meant to be delivered to every participant in the IG. Instead, most participants received an individual intervention, fewer than 25% were referred to community resources, and only a few attended group sessions. An intervention in which every participant receives the same complete intervention may lead to better outcomes. Furthermore, strategies for behavioral modification may lead to better results if they include goal-setting, self-monitoring, feedback, consequences, exercise prescriptions, and cues to action. Our study, which was based on the Transtheoretical Model, aimed to determine which participants were willing to change and, therefore, eligible to receive the PA intervention. However, this approach seemed to have only a small effect and may not be the optimal approach to the problem of inadequate PA [52].

A growing number of studies have examined the effect of different interventions for PA promotion in health care settings, but adopting a model of disease prevention rather than focusing on treatment and cure, requires a change in perspective. On the one hand, health professionals should provide individuals with information regarding the importance of PA for the prevention and management of NCD. Delivering consistent information on the performance of PA as a standard clinical practice may require specific training of health professionals and the acquisition of new skills. However, their lack of time, work overload, limited competencies, and specific beliefs may hinder their acceptance of performing regular PA counseling [53,54]. Our results also reflected the difficulties of providing referral or advice, as indicated by the large number of participants who were eligible but did not get a prescription for PA from their PHC professionals.

### Strengths and Limitations

The strength of this study is that it used a pragmatic approach to evaluate the effectiveness of an intervention in real-world conditions, and it was implemented by PHC professionals in several regions of Spain. We also included participants with two or more unhealthy behaviors and who therefore had a greater risk for poor health. Our MHBC intervention had several different components (individual sessions, group sessions, and community referral) to target several of the most common unhealthy habits simultaneously. This was designed as an efficient and original approach to address the growing burden of NCD.

This study also had some limitations. Because this was a cluster-randomized trial, we could not exclude the heterogeneity of the two groups. Thus, participants in the IG group may have been more accepting of making behavioral changes and engaging in PA than those in the CG. Although we measured PA levels using a validated questionnaire, the results were based on self-reported data. The use of additional objective methods for determining changes in PA would have provided useful information. Only part of the IG with risk of insufficient PA received the intervention, and it varied among participants. Moreover, the participants’ stage of change based on TTM was only recorded at baseline and at 12 months in the intervention group. In contrast to recommendations regarding diet and tobacco cessation, PA prescription is still quite novel, and there is no general agreement on the best approach for PHC settings. Providing advice on PA is outside the expertise of many GPs and might seem less important than other unhealthy behaviors, such as smoking. Nurses and physiotherapists are generally more confident and have more positive attitudes towards PA prescription, although all the health professionals who participated in this study received the same training before study onset. Still, this limitation might have negatively influenced the prescription of PA [55,56,57].

The reduced effectiveness of interventions that target multiple health behaviors could be due to a dilution effect over time of advice about PA or because participants initially engaged in other health behaviors (e.g., dietary changes or adherence to a Mediterranean diet) but not PA. This reduced effectiveness of interventions was also more evident because all patients in the MHBC intervention group were evaluated regardless of their engagement in any particular health behavior.

It may also have made it difficult to evaluate the effectiveness of the intervention and to provide recommendations or a reference for future integration of PA prescriptions in health care centers.

## 5. Conclusions

The MHBC intervention described here increased the proportion of participants who achieved the minimum PA recommended by the WHO (150 min/week of moderate PA), but did not lead to significant improvements in the promotion of PA nor in clinical biomarkers at 12 months.

The use of MHBC interventions at the PHC level to promote healthy lifestyles may help to prevent and manage NCDs. However, future studies are needed to assess the best strategies and the impact of different MHBC interventions in the promotion of healthy lifestyles and the mechanisms and contexts in which the interventions can produce the greatest benefits.

## Figures and Tables

**Figure 1 ijerph-18-10537-f001:**
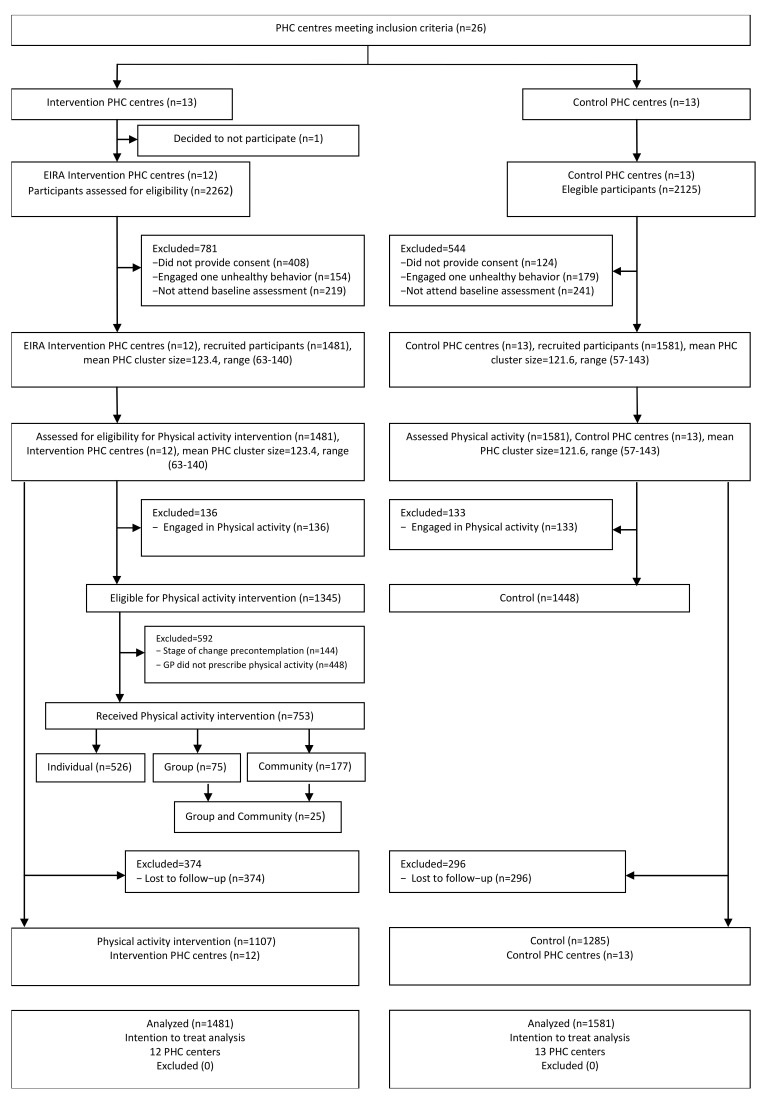
Enrolment and disposition of participants in the intervention and control groups.

**Table 1 ijerph-18-10537-t001:** Baseline characteristics of participants in the control and intervention group.

Basal Characteristics of the Participants	Control (*n* = 1581)	Intervention (*n* = 1481)
Sex, n/N (%)		
Men	709/1581 (44.8%)	672/1481 (45.4%)
Women	872/1581 (55.2%)	809/1481 (54.6%)
Age (years), mean ± SD	58.3 ± 8.3	57.7 ± 7.9
(mean 95% CI)	(57.9, 58.7)	(57.3, 58.1)
Civil status, n/N (%)		
Married/Co-habitant	1.055/1575 (67.0%)	1024/1461 (70.1%)
Single/Living apart	520/1575 (33%)	437/1461 (29.9%)
Work status		
Active	713/1575 (45.3%)	661/1456 (45.4%)
Inactive	431/1575 (27.4%)	424/1456 (29.1%)
Retired	431/1575 (27.4%)	371/1456 (25.5%)
Level of studies		
Higher education, university or similar	267/1572 (17.0%)	247/1461 (16.9%)
High school	621/1572(39.5%)	573/1461 (39.2%)
Primary school, not finished primary school, or without studies	684/1572 (43.5%)	641/1461 (43.9%)
Weight (kg), mean ± SD	78.77 ± 17.41	82.3 ± 17.81
(mean 95% CI)	(77.91, 79.63)	(81.38, 83.21)
BMI (kg/m^2^), mean ± SD	29.3 ± 5.6	30.6 ± 5.9
(mean 95% CI)	(29.0, 29.5)	(30.3, 30.9)
BMI, n/N (%)		
Normal	363/1574 (23.1%)	228/1454 (15.7%)
Overweight	595/1574 (37.8%)	491/1454 (33.8%)
Obese	616/1574 (39.1%)	735/1454 (50.6%)
Waist circumference (cm), mean ± SD	99.9 ± 15.0	102.0 ± 14.4
(mean 95% CI)	(99.1, 100.6)	(101.2, 102.7)
Physical activity level (MET mins/week) mean ± SD	1794.2 ± 3018.0	1480.0 ± 2382.0
(mean 95% CI)	(1644.2, 19442.2)	(1356.7, 1306.3)
Physical activity level, n/N (%)		
Low	709/1557 (45.5%)	729/1437 (50.7%)
Moderate	648/1557 (41.6%)	570/1437 (39.7%)
High	200/1557 (12.8%)	138/1437 (9.6%)
Follow WHO recommendations (Yes), n/N (%)	252/1580 (15.9%)	205/1476 (13.9%)
Moderate-intensity physical activity (min/day) mean ± SD	20.1 ± 56.4	16.4 ± 51.5
(mean 95% CI)	(17.2, 22.9)	(13.7, 19.1)
High-intensity physical activity (min/day) mean ± SD	9.2 ± 45.9	7.6 ± 36.1
(mean 95% CI)	(6.9, 11.6)	(5.7, 9.5)
Time spent sitting (min/day) mean ± SD	328.7 ± 188.7	321.2 ± 176.9
(mean 95% CI)	(319.2, 338.1)	(311.9, 330.4)
Stages of change: physical activity, n/N (%)		
Precontemplation	643/1548 (41.5%)	152/1270 (12.0%)
Contemplation	389/1548 (25.1%)	229/1270 (18.0%)
Preparation	204/1548 (13.2%)	516/1270 (40.6%)
Action	137/1548 (8.9%)	266/1270 (20.9%)
Maintenance/Termination	175/1548 (11.3%)	107/1270 (8.4%)
Diet, n/N (%)		
Inadequate	1261/1576 (80.0%)	1240/1480 (83.8%)
Adequate	315/1576 (20.0%)	240/1480 (16.2%)
Social support DUKE-UNC-11 mean ± SD	45.5 ± 8.4	45.4 ± 8.9
(mean 95% CI)	(45.0, 45.9)	(44.9, 45.9)
Health status EQ5D5L mean ± SD	67.6 ± 20.0	65.0 ± 21.0
(mean 95% CI)	(66.6, 68.6)	(63.9, 66.1)
Depression (Yes) n/N (%)	211/1196 (17.6%)	219/1140 (19.2%)
Tobacco use, n/N (%)		
Non-smoker	444/1581 (28.1%)	432/1481 (29.2%)
Ex-smoker	440/1581 (27.8%)	408/1481 (27.5%)
Smoker	697/1581 (44.1%)	641/1481 (43.3%)
Diabetes (Yes), n/N (%)	323/1579 (20.5%)	277/1469 (18.9%)
Systolic blood pressure (mmHg) mean ± SD	130.8 ± 17.5	132.85 ± 16.5
(mean 95% CI)	(129.8, 131.8)	(131.8, 133.9)
Diastolic blood pressure (mmHg) mean ± SD	79.88 ± 10.31	82.27 ± 10.08
(mean 95% CI)	(79.3, 80.5)	(81.6, 82.9)
Hypertension (Yes), n/N (%)	610/1577 (38.7%)	587/1459 (40.2%)
LDL mean ± SD	108.3 ± 33.9	108.0 ± 39.3
(mean 95% CI)	(104.3, 112.3)	(103.2, 112.8)
HDL mean ± SD	45.9 ± 12.2	46.7 ± 12.8
(mean 95% CI)	(44.5, 47.4)	(45.14, 48.28)
TG mean ± SD	131.4 ± 81.7	135.1 ± 80.1
(mean 95% CI)	(126.2, 136.7)	(129.6, 140.5)
Blood sugar mean ± SD	102.1 ± 29.6	104.3 ± 31.8
(mean 95% CI)	(100.2, 103.9)	(102.2, 106.5)
Glycated hemoglobin mean ± SD	7.19 ± 1.37	7.12 ± 1.36
(mean 95% CI)	(7.03, 7.35)	(6.95, 7.29)
Use of statin (Yes), n/N (%)	420/1569 (26.8%)	449/1462 (30.7%)
Cardiovascular disease (Yes), n/N (%)	87/1568 (5.5%)	87/1452 (6%)
REGICOR Cardiovascular risk mean ± SD	5.4 ± 0.1	5.6 ± 0.1
(mean 95% CI)	(5.2, 5.6)	(5.4, 5.7)

SD: standard deviation; 95% CI: 95% interval of confidence; BMI: Body Mass index; MET: metabolic equivalent of task minutes; WHO: World Health Organization; EQ5D5L: 5-level EQ-5D version questionnaire; REGICOR: cardiovascular risk calculator; LDL: low-density lipoprotein; HDL: high-density lipoprotein. TG: triglycerides.

**Table 2 ijerph-18-10537-t002:** Effect of the multiple health behavior change intervention based on per-protocol and intention-to-treat analysis.

	Baseline	At 12 Months			Per-Protocol		Intention to Treat	
	Control (*n* = 1581)	Intervention (*n* = 1481)	Control (*n* = 1285)	Intervention (*n* = 1107)	Mean Difference (95% CI)	*p*-Value	Mean Difference (95% CI)	*p*-Value
MET mins/week mean ± SD	1794.2 ± 3018.0	1480.0 ± 2382.0	1508.1 ± 2135.3	1507.1 ± 1996.0	52.37	0.534	284.09	0.335
(mean 95% CI)	(1644.2–19442.2)	(1356.7–1306.3)	(1387.7, 1628.4)	(1385.0, 1629.2)	(−112.61, 217.34)		(−298.24, 866.42)	
					OR (95% CI)	*p*-Value	OR (95% CI)	*p*-Value
Physical activity (IPAQ7)						0.800		0.822
Low	709/1557 (45.5%)	729/1437 (50.7%)	505/1220 (41.4%)	473/1082 (43.7%)	Ref		Ref	
Moderate-High	848/1557 (55.5%)	708/1437 (49.3%)	715/1220 (58.6%)	609/1082 (56.3%)	0.98 (0.82, 1.16)		1.02 (0.85, 1.23)	
Follow WHO recommendations								0.020
No	1328/1580 (84.1%)	1271/1476 (86,1%)	1059/1231 (86.0%)	906/1086 (83.4%)	ref		ref	
Yes	252/1580 (15.9%)	205/1476 (13.9%)	172/1231 (14.0%)	180/1086 (16.6%)	1.33 (1.05, 1.70)	0.019	1.29 (1.04, 1.60)	
					Mean Difference (95% CI)	*p*-Value	Mean Difference (95% CI)	*p*-Value
BMI (kg/m^2^) mean ± SD	29.3 ± 5.6	30.6 ± 5.9	28.97 ± 5.50	30.29 ± 5.65	−0.11	0.134	0.03	0.961
(mean 95% CI)	(29.0–29.5)	(30.3–30.9)	(28.66, 29.28)	(29.94, 30.63)	(−0.25, 0.03)		(−1.22, 1.28)	
Weight (kg) mean ± SD	78.77 ± 17.41	82.3 ± 17.81	78.25 ± 17.31	81.36 ± 17.34	−0.30	0.097	−0.04	0.982
(mean 95% CI)	(77.91–79.63)	(81.38–83.21)	(77.28, 79.22)	(80.31, 82.41)	(−0.66, 0.05)		(−3.30, 3.22)	
Waist circumference (cm) mean ± SD	99.9 ± 15.0	102.0 ± 14.4	98.67 ± 14.80	101.43 ± 14.20	0.60	0.043	1.10	0.454
(mean 95% CI)	(99.1–100.6)	(101.2–102.7)	(97.82, 99.52)	(100.55, 102.31)	(0.02, 1.18)		(−1.81, 4.00)	
Systolic BP (mmHg) mean ± SD	130.8 ± 17.5	132.85 ± 16.5	131.81 ± 17.14	131.00 ± 16.42	−1.96	0.001	−1.13	0.497
(mean 95% CI)	(129.8–131.8)	(131.8–133.9)	(130.84, 132.78)	(130.00, 132.01)	(−3.13, −0.79)		(−4.41, 2.16)	
Diastolic BP (mmHg) mean ± SD	79.88 ± 10.31	82.27 ± 10.08	80.07 ± 10.26	81.07 ± 9.94	−0.13	0.737	0.30	0.779
(mean 95% CI)	(79.3–80.5)	(81.6–82.9)	(79.49, 80.65)	(80.46, 81.68)	(−0.89, 0.63)		(−1.80, 2.41)	
LDL(mg/dL) mean ± SD	108.3 ± 33.9	108.0 ± 39.3	105.75 ± 32.51	104.63 ± 32.09	−1.92	0.165	−2.49	0.530
(mean 95% CI)	(104.3–112.3)	(103.2–112.8)	(100.98–110.51)	(99.80–109.46)	(−4.63, 0.79)		(−10.38, 5.39)	
HDL(mg/dL) mean ± SD	45.9 ± 12.2	46.7 ± 12.8	47.76 ± 12.85	47.28 ± 13.23	−0.08	0.828	−0.28	0.843
(mean 95% CI)	(44.5–47.4)	(45.14–48.28)	(45.87, 49.64)	(45.29, 49.27)	(−0.85, 0.68)		(−3.08, 2.52)	
TG (mg/dL) mean ± SD	131.4 ± 81.7	135.1 ± 80.1	130.9 ± 81.7	138.5 ± 81.6	5.41	0.051	7.99	0.487
(mean 95% CI)	(126.2–136.7)	(129.6–140.5)	(126.5, 135.3)	(132.9, 144.0)	(−0.02, 10.84)		(−14.76, 30.73)	
Blood sugar (mg/dL) mean ± SD	102.1 ± 29.6	104.3 ± 31.8	135.94 ± 46.84	135.36 ± 38.75	−0.72	0.491	0.88	0.763
(mean 95% CI)	(100.2–103.9)	(102.2–106.5)	(129.07–142.81)	(129.52–141.19)	(−2.78, 1.34)		(−4.91, 6.68)	
Glycated hemoglobin (%) mean ± SD	7.19 ± 1.37	7.12 ± 1.36	6.97 ± 1.15	6.81 ± 1.05	−0.10	0.273	−0.08	0.719
(mean 95% CI)	(7.03–7.35)	(6.95–7.29)	(6.80–7.14)	(6.65–6.97)	(−0.28, 0.08)		(−0.51, 0.35)	
REGICOR mean ± SD	5.4 ± 0.1	5.6 ± 0.1	5.33 ± 3.36	5.34 ± 3.60	−0.26	0.030	−0.15	0.464
(mean 95% CI)	(5.2–5.6)	(5.4–5.7)	(5.10, 5.55)	(5.08, 5.59)	(−0.50, −0.03)		(−0.56, 0.26)	
Moderate-intensity physical activity (IPAQ7)						0.038		0.100
(min/day) mean ± SD	20.1 ± 56.4	16.4 ± 51.5	16.15 ± 48.73	18.54 ± 52.23	0.25		0.21	
(mean 95% CI)	(17.2–22.9)	(13.7–19.1)	(13.41, 18.88)	(15.40, 21.68)	(0.01, 0.49)		(−0.04, 0.45)	
High-intensity physical activity (IPAQ7)						0.491		0.402
(min/day) mean ± SD	9.2 ± 45.9	7.6 ± 36.1	6.60 ± 36.61	7.08 ± 35.13	0.16		0.15	
(mean 95% CI)	(6.9–11.6)	(5.7–9.5)	(4.55, 8.66)	(4.97, 9.19)	(−0.29, 0.60)		(−0.20, 0.49)	
Mins/day spent sitting (IPAQ7)						0.764		0.920
mean ± SD	328.7 ± 188.7	321.2 ± 176.9	314.66 ± 160.94	309.48 ± 165.54	0.01		−0.003	
(mean 95% CI)	(319.2–338.1)	(311.9–330.4)	(305.62, 323.69)	(299.54, 319.43)	(−0.03, 0.04)		(−0.07, 0.06)	

## Data Availability

The data that support the findings of this study are available on reasonable request from the corresponding author at aleiva@ibsalut.es. Proposals requesting data access will need to specify how it is planned to use the data.

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
