# Peer review of "Implementation of the EIRA 3 Intervention by Targeting Primary Health Care Practitioners: Effectiveness in Increasing Physical Activity"

_ijerph, 2021, doi:10.3390/ijerph181910537_

Round 1

Reviewer 1 Report

Thank you for taking the time to make the significant revisions requested. Overall, the paper appears to contain some interesting data, but the intervention details are still confusing, and the presentation of results is cumbersome. Below are some specific comments for consideration.

  • The eligibility criteria are confusing. Is my interpretation correct in that people who were physically active were eligible for the intervention? This doesn’t seem appropriate given the study purpose.
  • The intervention itself is confusing. What was the difference between the very brief and the brief intervention? How many received each? Did the results vary? More detail is needed on the full intervention, not just the activity components.
  • The sample size calculation seems problematic. The authors only used 20% power for their calculation? The effect size is also quite low. More clarity is needed on how the sample size was calculated, and what the power and effect sizes were in the current study.
  • The control group had significantly higher physical activity levels at baseline. Was this accounted for in the analysis and interpretation of health outcomes?
  • In table 2, are the data presented from end of intervention? It would be useful to see the pre and post data side by side for both groups to understand the effect of the intervention. Furthermore, effect sizes are needed.

Author Response

Dear Editor,

Thank you for giving me the opportunity to submit a revised draft of my manuscript titled “Implementation of the EIRA 3 intervention by targeting primary health care practitioners: Effectiveness in increasing physical activity”.

We want to thanks the reviewer for the revision of the manuscript and the opportunity to improve and resubmit the manuscript.

The following constitutes a point-by-point response to the comments of the reviewer.

Thank you for your consideration of this manuscript.

Sincerely,

Alfonso Leiva

REVIER’S COMMENTS

Thank you for taking the time to make the significant revisions requested. Overall, the paper appears to contain some interesting data, but the intervention details are still confusing, and the presentation of results is cumbersome. Below are some specific comments for consideration.

  • The eligibility criteria are confusing. Is my interpretation correct in that people who were physically active were eligible for the intervention? This doesn’t seem appropriate given the study purpose.

Thanks the reviewer for the insightful comments. The study was designed as a RCT to evaluate the effectiveness of a multiple risk intervention, participants were included if they had at least 2 of the 3 target unhealthy behaviours (tobacco use, physical inactivity and reduced fruit and vegetable consumption). Although most of the participants were inactive

“A total of 3062 participants from the 25 PHC centers met eligibility criteria. The IG had 1481 participants, and we classified 1345 of them (90.8%) as inactive, based on the Brief Physical Activity Assessment Tool.”

There were some patients that were eligible for reduced fruit and vegetable consumption and tobacco use and not for physical inactivity (9,2%)

We identified as a limitation of the study and it was discussed in the limitations section of the manuscript.

“The reduced effectiveness of interventions that target multiple health behaviors could be due to a dilution effect over time of advice about PA or because participants initially engaged in other health behaviors (e.g. dietary changes or adherence to a Mediterranean diet) but not PA. This reduced effectiveness of interventions was also more evident because all patients in the MHBC intervention group were evaluated regardless of their engagement in any particular health behavior.“)

  • The intervention itself is confusing. What was the difference between the very brief and the brief intervention?

I would like to thank reviewer  for drawing my attention to the lack of clarity in this section of the manuscript. The multiple risk intervention was based on the Transtheoretical Model of behaviour change  and those patients in precontemplation or contemplation stage received a brief motivational intervention to promote PA, and patients in the preparation or action stage received an “action plan” that include the prescription of PA in individual, or group session. We are aware of the reviewer concern about the description of the intervention and we have added an additional supplementary table and try to make clearer the description of the different components of the intervention.

“After evaluating the state of change, the PHC recommended either a) a “very brief intervention” with the aim of increasing awareness and highlighting the importance of the need for behavior change or to support the change and help with relapse prevention; or b) a “brief intervention” which also included the formulation and agreement of an action plan for behavior change, preferably PA involving group sessions and community resources offered by the EIRA study”

...How many received each? Did the results vary?

Although this would have been of high interest, this information was not recorded in the study.

More detail is needed on the full intervention, not just the activity components.

In order to provide additional information regarding the intervention we have included a supplementary table (table S1) and some extra details to clarify some aspects.

  • The sample size calculation seems problematic. The authors only used 20% power for their calculation?

Please note the difference between power and beta risk.

 β = probability of a Type II error, known as a "false negative"

1-β = probability of a "true positive", which is correctly rejecting the null hypothesis. "1-β" is also known as the power of the test.

 We have 80% probability (1-beta risk) of correctly rejecting the Null Hypothesis. We believe that we have enough power to detect clinically meaningful effects of the intervention.

“The sample size was calculated so that the study was sufficiently powered to detect differences in the proportion of participants who changed at least one unhealthy behavior [20]. A total of 3640 participants were initially recruited and 3062 participants were finally included. The standard deviation of the MET mins/week in the study population was 2700. Assuming an alpha risk of 5%, a beta risk of 20%, and an intra-cluster correlation of 0.01 [40], this allowed detection of an effect size of 0.26 (Cohen’s d) in PA measured by MET mins/week difference between the control and intervention groups.”

The effect size is also quite low. More clarity is needed on how the sample size was calculated, and…. 

As the reviewer stated, the study sample size allows us to detect clinically meaningful effects of the intervention (effect sizes) even if these effects were low, a Cohen´s d 0,26 means that we could detect differences of 0,26 standard deviations of the main outcomes.

We followed the consort recommendation to report sample size and we believe the sample size is reported clearly.

“For scientific and ethical reasons, the sample size for a trial needs to be planned carefully, with a balance between medical and statistical considerations. Ideally, a study should be large enough to have a high probability (power) of detecting as statistically significant a clinically important difference of a given size if such a difference exists. The size of effect deemed important is inversely related to the sample size necessary to detect it; that is, large samples are necessary to detect small differences. Elements of the sample size calculation are (1) the estimated outcomes in each group (which implies the clinically important target difference between the intervention groups); (2) the α (type I) error level; (3) the statistical power (or the β (type II) error level); and (4), for continuous outcomes, the standard deviation of the measurements.” (Altman DG, Schulz KF, Moher D, Egger M, Davidoff F, Elbourne D, Gøtzsche PC, Lang T; CONSORT GROUP (Consolidated Standards of Reporting Trials). The revised CONSORT statement for reporting randomized trials: explanation and elaboration. Ann Intern Med. 2001 Apr 17;134(8):663-94.)

...what the power and effect sizes were in the current study.

 We have 80% power to detect 0,26 standard deviations of Physical activity energy expenditure differences between intervention and control groups.

  • The control group had significantly higher physical activity levels at baseline. Was this accounted for in the analysis and interpretation of health outcomes?

Thank the reviewer for the comment. All analysis were adjusted for the baseline values. We agree that is not clear enough and we add a sentence in the result section to clarify:

 “Differences in adjusted mean change from baseline between intervention versus control at 12 months are presented in Table 2”

  • In table 2, are the data presented from the end of intervention? It would be useful to see the pre and post data side by side for both groups to understand the effect of the intervention. Furthermore, effect sizes are needed.

Thank the reviewer. We have changed Table 2 and Table 3 to present baseline and follow up values in the same table, effect sizes are presented as adjusted mean differences between intervention and control groups.

Reviewer 2 Report

The authors have revised appropriately as requested by this reviewer.

Author Response

The authors have revised appropriately as requested by this reviewer.

We want to thanks the reviewer for the revision of the manuscript 

This manuscript is a resubmission of an earlier submission. The following is a list of the peer review reports and author responses from that submission.

Round 1

Reviewer 1 Report

This is an interesting intervention study conducted in a large sample of adults. Clarification about several aspects of the study are necessary prior to publication.

Generally, the methods section requires significant improvements with regards to detail and flow. I would suggest carefully following the CONSORT statement to ensure sufficient detail is provided throughout.

It is unclear what was used for the primary outcome of the study. PA measures are mentioned in the participant section and again in the outcome section. Which was used for analysis? Why were multiple measures used. What is the validity and reliability of the first measure? More detail is needed to understand which measures were used and what cut points were used for the various analyses.

Why were BMI and BP used as outcomes in this study? The purpose statement mentioned only physical activity, therefore, more background information, and a more detailed purpose statement need to be included for inclusion of these outcomes.

Details on the intervention itself are also sparse. What was the EVIDENT app used for, what messages were sent via SMS, what conversations were conducted in groups. Please include as much detail as possible regarding the intervention so that the reader can understand why PA was not affected. Were tobacco and dietary interventions provided simultaneously? What were the differences between the individual, group and community interventions. This is important information for the reader.

Minor point: It is recommended to use ‘participants’ instead of ‘subjects’.

Reviewer 2 Report

Understanding how to support healthy behavior change, particularly increases in physical activity is important. Targeting primary healthcare practitioners is also important.  Several revisions would assist in the presentation of these results.

1) How was the dilution of targeting physical activity as a MHBC intervention to target accounted for in the analyses of the results?

2) the authors state that the intervention is  personalized based on targeted behaviors (in this case PA) and their stage of change as identified by the Transtheoretical Model yet it is difficult to evaluated how this was used to guide the analysis which does not seem to evaluate any within individual changes in the intervention arm.

3) Table 1 presents the stage of change for PA at the beginning of the study but did this change over time during their engagement with the PHC professional in the intervention particularly since there was a significant change seen in their knowledge of the WHO recommendations?

4) were there other biomarkers collected that were not presented in the table? 

5) The authors acknowledge the heterogeneity of the sites, availability of community resources, number and types of interventions (individual, group, community referral but non of these are presented in the analysis. Without any of this information even at a qualitative level the conclusions are difficult to interpret.